# REVISITING INSTANCE-REWEIGHTED ADVERSARIAL TRAINING

## ABSTRACT

Instance-reweighted adversarial training (IRAT) is a type of adversarial training that assigns large weights to high-importance examples and then minimizes the weighted loss. The importance often uses the margins between decision boundaries and each example. In particular, IRAT can alleviate robust overfitting and obtain excellent robustness by computing margins with an estimated probability. However, previous works implicitly dealt with binary classification even in the multi-class cases, because they computed margins with only the true class and the most confusing class. The computed margins can become equal even with different true probability examples, because of the complex decision boundaries in multi-class classification. In this paper, first, we clarify the above problem with a specific example. Then, we propose *margin reweighting*, which can transform the previous margins into appropriate representations for multi-class classification by leveraging the relations between the most confusing class and other classes. Experimental results on the CIFAR-10/100 datasets demonstrate that the proposed method is effective in boosting the robustness against several attacks as compared to the previous methods.

## 1    INTRODUCTION

While convolutional neural networks (CNNs) achieve excellent performance on various tasks, they are vulnerable to adversarial examples (Szegedy et al., 2014; Goodfellow et al., 2015) with malicious, perturbed samples. Such perturbation is a threat against CNN-based AI systems (e.g., those for autonomous driving or medical diagnosis) because it is imperceptible to humans. Thus, there are various effective approaches to mitigate the negative impact of perturbation (Papernot et al., 2016; Samangouei et al., 2018; Xu et al., 2018; Madry et al., 2018). Among them, adversarial training (AT) (Madry et al., 2018) is well known as an attractive defense strategy because of its clarity and efficacy.

Instead of using benign examples, AT trains adversarial examples generated by projected gradient descent (PGD) (Madry et al., 2018). Although AT can achieve excellent robustness, it can also exhibit performance degradation for benign examples or robust overfitting (Zhang et al., 2019; Rice et al., 2020). Instance-reweighted adversarial training (IRAT) (Zeng et al., 2021; Kim et al., 2021; Zhang et al., 2021; Wang et al., 2021; Gao et al., 2021) is an effective method among the many approaches developed to overcome these issues.

IRAT computes the margins between the decision boundaries and each example as the importance, which is transformed into weights with a nonlinear increasing function. Then, it minimizes the weighted classification loss by assigning these weights to each example. Geometry-aware instance-reweighted adversarial training (GAIRAT), proposed by (Zhang et al., 2021), decides the importance for each example by the least PGD steps (LPS). GAIRAT represents the margin in an input space, because the LPS is the number of steps to cause an adversarial example to cross decision boundaries, starting from a benign example. Smaller-margin examples are closer to the decision boundaries and are assigned larger weights. Although GAIRAT achieves better robustness than standard AT, it is vulnerable against attacks other than PGD because it defines the importance in terms of the LPS.

Meanwhile, margin-aware instance reweighting learning (MAIL) (Wang et al., 2021) successfully overcomes the weakness of GAIRAT by defining the importance with estimated probabilities. Specifically, it transforms the difference between two probabilities, i.e., between the true class and

the most confusing class, to a weight with a nonlinear increasing function. Weighted minimax risk (WMMR) (Zeng et al., 2021) uses the same approach. The difference between these methods is the use of different weighting functions, and MAIL achieves better performance than WMMR. Unlike GAIRAT with its discrete representation for weights, MAIL alleviates robust overfitting by using continuous weights. However, MAIL and WMMR have a problem in that the importance cannot be adequately represented in multi-class classification, because only the most confusing class and the true class are considered. As shown in Fig. 1(b), we assume that this issue occurs in previous approaches for instances that have the same margin. Intuitively, among examples that have the same margin, such as $x_1$ and $x_2$, the importance should be large closer to the intersection of the decision boundaries of multiple classes, but previous methods neglect this representation.

In this paper, to resolve this issue, we reveal the problem with the previous margin computation through a specific example. Then, as illustrated in Fig. 1(c), we propose *margin reweighting*, which enables us to transform the previous margins to an appropriate representation by considering classes other than the most confusing class and the true class. Although there is a straightforward approach for this, which arises from computing the margins between the true class and other classes (i.e., by using a multi-class margin), it is hard to design a weighting function to aggregate the multi-class margin to a single weight. Thus, we propose a novel metric: the ratio of the top2 in the incorrect rate (i.e., the sum of all probabilities except the true class). Assuming that each class probability well represents relationship to the center of a class, this metric can identify whether examples are close to the intersections of multi-class boundaries. Therefore, we do not have to design a special weighting function, and we can get appropriate representations just by multiplying this measure by the previous margins. We performed experiments and demonstrated that the proposed method can boost the performance of the previous methods for certain attacks. In summary, our work makes the following contributions:

- We clarify that the previous approach of computing the margin by using predicted probabilities is insufficient. Specifically, we show a case in which the same margins are computed for certain examples even though both the true and most confusing class probabilities are different.

- We propose *margin reweighting*, which enables transformation of the previous margins into appropriate ones by leveraging a relation between the most confusing class probability and the incorrect rate.

- We experimentally show that our approach is effective for boosting the robustness against adversarial attacks.

## 2 PRELIMINARIES AND RELATED WORKS

In this section, we give an overview of standard training and adversarial training (AT), and then we describe related works on instance-reweighted adversarial training (IRAT).

### 2.1 STANDARD TRAINING VS. ADVERSARIAL TRAINING

**Standard training:** Let $\mathcal{D} = \{x_i, y_i\}_{i=1}^n$ be a training dataset where $x_i \in \mathbb{R}^{c \times h \times w}$ is an input example and $y_i$ is a ground truth label. In standard training, a deep neural net $f : \mathbb{R}^{c \times h \times w} \rightarrow \mathbb{R}^K$ parameterized by $\theta$ minimizes the loss $\ell(f_\theta(x_i), y_i)$:

$$\min_\theta \mathbb{E}_{(x_i, y_i) \sim \mathcal{D}}[\ell(f_\theta(x_i), y_i)], \tag{1}$$

where $K$ is the number of classes, and the loss function $\ell(\cdot)$ uses the cross-entropy loss.

**Adversarial training:** AT (Madry et al., 2018) aims to obtain a model with robustness against adversarial attacks by training on computed adversarial examples. Actually, in the field except for image classification, this framework uses before deep learning became popular (Dalvi et al., 2004; Lowd & Meek, 2005). Unlike standard training, AT consists of two processes, inner maximization and outer minimization. First, the inner maximization computes a perturbation $\delta_i$ so as to maximize the loss within a radius $\epsilon$ centered on $x_i$. Then, the outer minimization updates the weight parameters $\theta$ to minimize the loss with adversarial examples $\hat{x}_i = x_i + \delta_i$ obtained by the inner maximization.

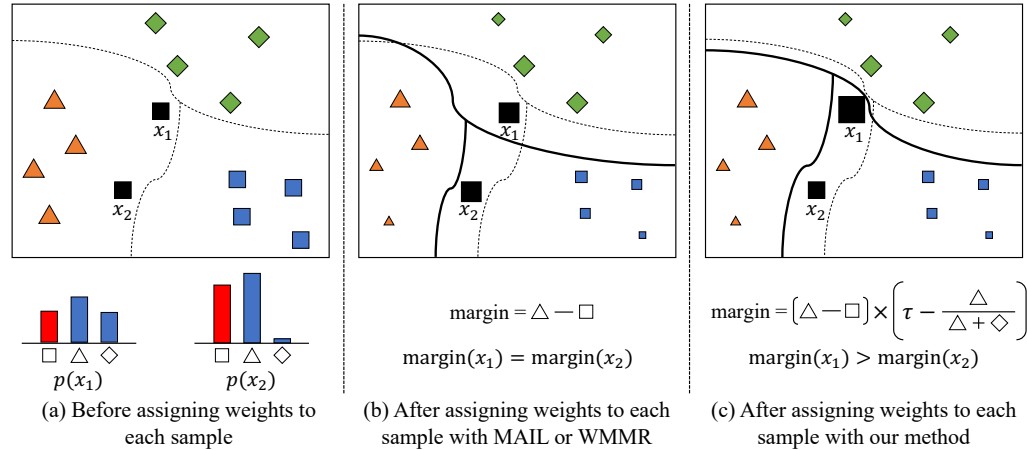

Figure 1: Differences between both previous methods (i.e., MAIL and WMMR) and our method. The size of each shape in (b) and (c) indicates the amount of weight, and the solid and dashed lines represent the decision boundaries before and after updates, respectively. Each shape represents a different class. (b) The previous methods assign equal weights to equal margins because they only use the difference between probabilities of the true class and the most confusing class. However, we should define a larger weight for $x_1$ than for $x_2$, because $x_1$ is close to the intersection of multiple class boundaries. Thus, (c) our method can transform the margins for $x_1$ and $x_2$ to appropriate forms in multi-class classification.

The following equation represents these processes:

$$\min_{\boldsymbol{\theta}} \mathbb{E}_{(\boldsymbol{x}_i, y_i) \sim \mathcal{D}} \left[ \max_{\|\boldsymbol{\delta}_i\|_p \leq \epsilon} \ell(f_{\boldsymbol{\theta}}(\boldsymbol{x}_i + \boldsymbol{\delta}_i), y_i) \right], \tag{2}$$

where $p = \{0, 1, 2, \infty\}$; we use $p = \infty$ in this paper.

The inner maximization often computes adversarial examples via a multi-step attack (e.g., PGD (Madry et al., 2018) with a step size $\alpha \leq \epsilon$):

$$\hat{\boldsymbol{x}}_i^{(t+1)} := \Pi_{\mathcal{B}[\boldsymbol{x}_i^{(0)}]} \left( \hat{\boldsymbol{x}}_i^{(t)} + \alpha \cdot \text{sign} \left( \nabla_{\hat{\boldsymbol{x}}_i^{(t)}} \ell(f_{\boldsymbol{\theta}}(\hat{\boldsymbol{x}}_i^{(t)}), y_i) \right) \right), \tag{3}$$

where $\mathcal{B}[\boldsymbol{x}_i^{(0)}] := \{\hat{\boldsymbol{x}}_i \in \mathcal{X} \mid \|\boldsymbol{x}_i^{(0)} - \hat{\boldsymbol{x}}_i\|_\infty \leq \epsilon\}$ in the input space $\mathcal{X}$, and $\Pi$ is a projection function that brings outliers into $\mathcal{B}[\boldsymbol{x}_i^{(0)}]$.

Although it is not hard to obtain robustness with AT, it has drawbacks of degrading the classification performance on benign examples and causing robust overfitting. Among various approaches to overcome these issues (Cai et al., 2018; Zhang et al., 2019; Uesato et al., 2019; Zhang & Wang, 2019; Wu et al., 2020; Wang et al., 2020; Zhang et al., 2020; Cheng et al., 2022; Ding et al., 2020; Song et al., 2021; Cui et al., 2021; Yu et al., 2022), we deal with IRAT (Zeng et al., 2021; Kim et al., 2021; Zhang et al., 2021; Wang et al., 2021; Gao et al., 2021).

## 2.2 INSTANCE-REWEIGHTED ADVERSARIAL TRAINING

IRAT is a form of adversarial training that defines weights based on the importance computed for each example and then minimizes the weighted loss. There are various IRAT approaches with different computations of importance.

Geometry-aware instance-reweighted adversarial training (GAIRAT) (Zhang et al., 2021) determines the importance of each example by using the least PGD steps (LPS). The LPS indicates the number of steps to cause an adversarial example to cross decision boundaries, starting from a benign instance. While GAIRAT can achieve excellent performance against a PGD attack, it is vulnerable to other strong attacks because the importance depends on the PGD attack of the inner maximization. To overcome this weakness, Gao et al. (2021) proposed to consider other attacks, such as Carlini-Wagner (CW) (Carlini & Wagner, 2017) or AutoAttack (AA) (Croce & Hein, 2020),

in the inner maximization; however, this approach is costly. Meanwhile, entropy-weighted adversarial training (EWAT) (Kim et al., 2021) improves the robustness against CW (Carlini & Wagner, 2017) and AutoAttack (AA) (Croce & Hein, 2020) by assigning an entropy to the input as a weight for each example.

In contrast to EWAT, weighted minimax risk (WMMR) (Zeng et al., 2021) and margin-aware instance reweighting learning (MAIL) (Wang et al., 2021) define margins for each example using the true class and the most confusing class probabilities, via the following equation:

$$m(\boldsymbol{x}_i, y_i) = \arg\max_{j \neq y_i} p_j(\boldsymbol{x}_i) - p_{y_i}(\boldsymbol{x}_i), \tag{4}$$

where $p_{y_i}(\boldsymbol{x}_i)$ indicates the true class probability, as $p(\boldsymbol{x}_i)$ is the predicted distribution. Note that these approaches use different weighting functions: sigmoid for MAIL, exponential for WMMR. MAIL resolves the weakness of GAIRAT and achieves better performance.

In this paper, we aim to define the margin with Eq. (4) as in the MAIL and WMMR. However, those methods do not give appropriate margins in multi-class classification, because they ignore relations other than those between the true class and the most confusing class. Hence, in the next section, we seek to address this problem.

## 3 PROPOSED METHOD: MARGIN REWEIGHTING

In this section, first, we reveal the weakness of the previous IRAT methods in multi-class classification. Specifically, we show that Eq. (4) may obtain the same margins with certain examples even though the probabilities of the true class and the most confusing class are different. Next, we propose a novel metric to represent relations between the most confusing class and other classes. Finally, we introduce weighting functions to incorporate our proposed method into the previous methods.

**Definition 1** (top2 probability). *Let $(\boldsymbol{x}_i, y_i) \sim \mathcal{D}$ be the input data and let $p(\boldsymbol{x}_i) \in [0, 1]^K$ be the posterior distribution for $\boldsymbol{x}_i$. Then, the top2 probability is defined as $p_2(\boldsymbol{x}_i) = \arg\max_{k \neq y_i} p_k(\boldsymbol{x}_i)$.*

**Remark:** Strictly speaking, the top2 probability is the the highest when $p_{y_i}(\boldsymbol{x}_i) < p_2(\boldsymbol{x}_i)$. In this paper, however, we call the top2 probability even in this situation.

### 3.1 WEAKNESS OF PREVIOUS IRAT METHODS

Although the previous IRAT methods with class probabilities achieve excellent performance, they do not account for classes other than the most confusing class. Except in easy cases, some samples typically lie near the intersections of multi-class decision boundaries. Thus, we can easily expect insufficient representations for WMMR and MAIL in multi-class classification.

**Lemma 1.** *For the margin calculated by Eq.(4) and a training dataset $\mathcal{D} := \{(\boldsymbol{x}_i, y_i)\}_{i=1}^n$, there exist samples that satisfy*

$$m(\boldsymbol{x}_i, y_i) = m(\boldsymbol{x}_j, y_j). \tag{5}$$

By applying Lemma 1, we examine the margins computed by Eq.(4). First, examples that have high confidence in terms of either the true class or the top2 probability are represented far from other classes. In other words, these examples are located around the center of either the true class or the top2 class. In this case, Eq.(5) does not hold true between

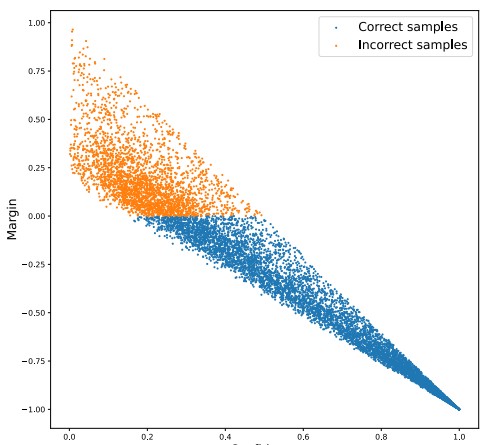

Figure 2: Correlation between the true class probabilities $p_{y_i}(\boldsymbol{x}_i)$ and the margins computed by Eq. 4, for ResNet-18 with standard adversarial training on the CIFAR-10 dataset. The orange and blue dots represent incorrect and correct samples, respectively.

high- and low-confidence examples, as illustrated in Fig. 2. Therefore, it is not a problem to define the margins with Eq.(4). On the other hand, low-confidence examples in both the true class and the top2 probability may be around the multi-class decision boundaries. Thus, these examples satisfy Eq.(5) not only for examples with both probabilities nearly equal but also for examples with

different probabilities (e.g., $p_{y_i}(\boldsymbol{x}_i) \ll p_{y_j}(\boldsymbol{x}_j)$). For instance, as shown in Fig. 2, we can obtain approximately equal margins for examples within [0.2, 0.6]. This represents the situation where $\boldsymbol{x}_i$ is closer to the intersection of multiple class boundaries and has a higher attack risk than $\boldsymbol{x}_j$. Hence, we argue that it is inappropriate to make the margins equal for these examples. Although we would expect improved robustness if we could carefully distinguish these margins, neither WMMR nor MAIL accounts for them.

### 3.2 QUANTIFICATION OF IMPORTANCE FOR MARGINS

The margin determination described above is insufficient when we recall that Lemma 1 holds for different probabilities among certain examples. A straightforward, intuitive approach to resolve this issue is to compute the margins between the true class and other classes with Eq. (4). However, that approach is hard to implement because it requires careful design of the weighting function to transform the multi-class margins to the appropriate weights.

**Definition 2** (Incorrect rate). *Let $(\boldsymbol{x}_i, y_i) \sim \mathcal{D}$ be the input data and let $p_{y_i}(\boldsymbol{x}_i) \in [0, 1]$ be the correct class probability. Then, the incorrect rate is defined as*

$$
\begin{aligned}
\bar{p}_{y_i}(\boldsymbol{x}_i) &= 1 - p_{y_i}(\boldsymbol{x}_i) \\
&= \sum\nolimits_{k \neq y_i} p_k(\boldsymbol{x}_i).
\end{aligned} \tag{6}
$$

Hence, we propose a metric that quantifies the relations between the most confusing class and other classes by using the top2 probability and the incorrect rate. Assuming that Lemma 1 holds for examples with different confidence, we can derive the following inequality for the ratio of the top2 probability to the incorrect rate.

**Proposition 1.** *For input data $(\boldsymbol{x}_i, y_i)$ and $(\boldsymbol{x}_j, y_j)$, let $m(\boldsymbol{x}_i, y_i) = m(\boldsymbol{x}_j, y_j)$, $p_{y_i}(\boldsymbol{x}_i) > p_{y_j}(\boldsymbol{x}_j)$, and $p_{y_j}(\boldsymbol{x}_j) > \frac{1}{K}$. Then, the following inequality for the ratio if the top2 probability to the incorrect rate holds for each sample:*

$$
o(\boldsymbol{x}_i, y_i) > o(\boldsymbol{x}_j, y_j), \quad \text{where } o(\boldsymbol{x}, y) = \frac{p_2(\boldsymbol{x})}{\bar{p}_y(\boldsymbol{x})}. \tag{7}
$$

The margins computed with Eq.(4) represent only the relation to decision boundaries or whether examples successfully classify. Meanwhile, for the examples with the same margin, Proposition 1 shows that they can further distinguish by using the ratio of the top2 probability included in the incorrect rate. According to MAIL and WMMR (i.e., that Eq.(4) implies the margin between the examples and the decision boundaries), we can determine that examples with a low ratio are distributed around the intersection of multiple classes. Thus, we expect that we can resolve our problem by applying the ratio of the top2 probability to the margin computed by Eq.(4).

Next, we discuss the range of the ratio $o(\boldsymbol{x}_i, y_i)$. Let $P_{\text{top2}} := \left( \frac{\bar{p}_{y_i}(\boldsymbol{x}_i)}{K-1}, \bar{p}_{y_i}(\boldsymbol{x}_i) \right)$ be the range of $p_2(\boldsymbol{x}_i)$; then, the upper and lower bounds obtained directly by using $o(\boldsymbol{x}_i, y_i)$ are defined as follows:

$$
\begin{cases}
\sup\limits_{p_2(\boldsymbol{x}_i) \in P_{\text{top2}}} o(\boldsymbol{x}_i, y_i) = 1 & p_2(\boldsymbol{x}_i) = \bar{p}_{y_i}(\boldsymbol{x}_i), \\
\inf\limits_{p_2(\boldsymbol{x}_i) \in P_{\text{top2}}} o(\boldsymbol{x}_i, y_i) = \frac{1}{K-1} & p_2(\boldsymbol{x}_i) = \frac{\bar{p}_{y_i}(\boldsymbol{x}_i)}{K-1}.
\end{cases} \tag{8}
$$

According to Eq.(8), direct use of the top2 ratio is contrary to our motivation, because we confirm that $o(\boldsymbol{x}_i, y_i)$ approaches the upper/lower bound when $p_2(\boldsymbol{x}_i)$ is large/small. Hence, we can obtain the appropriate range by inverting the top2 ratio with an arbitrary coefficient $\tau \in \mathbb{N}$. However, we cannot say that the representation is appropriate, because $\tau = 1$ gives the following upper and lower bounds:

$$
\begin{cases}
\sup\limits_{p_2(\boldsymbol{x}_i) \in P_{\text{top2}}} (\tau - o(\boldsymbol{x}_i, y_i)) = 1 - \frac{1}{K-1} & p_2(\boldsymbol{x}_i) = \frac{\bar{p}_{y_i}(\boldsymbol{x}_i)}{K-1}, \\
\inf\limits_{p_2(\boldsymbol{x}_i) \in P_{\text{top2}}} (\tau - o(\boldsymbol{x}_i, y_i)) = 0 & p_2(\boldsymbol{x}_i) = \bar{p}_{y_i}(\boldsymbol{x}_i).
\end{cases} \tag{9}
$$

Equation(9) implies that examples with $p_2(\boldsymbol{x}) \approx \bar{p}_y(\boldsymbol{x})$ (i.e., significantly misclassified adversarial examples) are no longer involved in the updating the parameters.

Hence, we define weights for the margins computed by Eq.(4) with the following equation:

$$
s(\boldsymbol{x}_i, y_i) = \tau - o(\boldsymbol{x}_i, y_i), \quad \text{s.t. } \tau \in \mathbb{N}_{\geq 2}, \tag{10}
$$

where $\mathbb{N}_{\geq 2}$ denotes the set of natural numbers greater than or equal to 2.

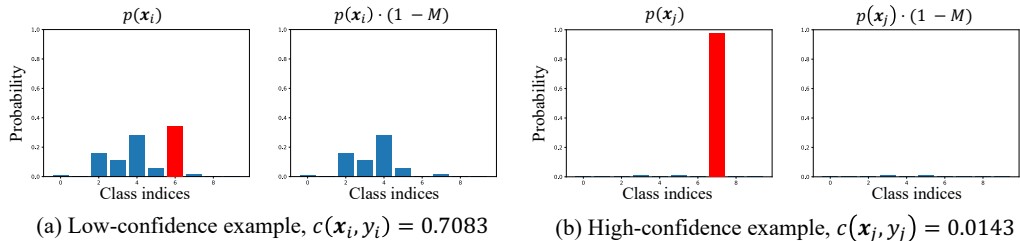

(a) Low-confidence example, $c(\boldsymbol{x}_i, y_i) = 0.7083$  (b) High-confidence example, $c(\boldsymbol{x}_j, y_j) = 0.0143$

Figure 3: Cosine similarity computed with Eq.(11) for the predicted distributions of two examples with (a) low and (b) high confidence. The red bar represents the true class in each example.

### 3.3 REPRESENTATION THE IMPORTANCE FOR THE INCORRECT RATE

As described above, we can represent whether examples are close to the intersection of multiple classes, but Eq.(10) is insufficient yet. Specifically, for both examples around the boundaries and successfully classified examples with high confidence, the top2 ratio may be equal if the incorrect class probabilities are uniformly equal. Because these examples require the definition of different weights, this implies that the importance of the incorrect rate is also different. Therefore, we represent the importance of the incorrect rate by leveraging the predicted probability distributions via the following equation for the cosine similarity:

$$c(\boldsymbol{x}_i, y_i) = \frac{< p(\boldsymbol{x}_i), p'(\boldsymbol{x}_i) >}{\|p(\boldsymbol{x}_i)\|_2 \cdot \|p'(\boldsymbol{x}_i)\|_2}. \tag{11}$$

Here, $p(\boldsymbol{x}_i) \in [0, 1]^K$ denotes the predicted probability distribution of $\boldsymbol{x}_i$. $p'(\boldsymbol{x}_i)$ denotes the distribution with elimination of the true class and computation using a binary mask, which has values of 1 except for the true class:

$$p'(\boldsymbol{x}_i) := p(\boldsymbol{x}_i) \cdot (1 - M), \ \ \text{s.t.} \ \sum_{k=1}^{K} p'_k(\boldsymbol{x}_i) < 1. \tag{12}$$

As shown in Fig. 3(a), the similarity tends to be large for an example with low confidence. Meanwhile, as shown in Fig. 3(b), it tends to be small for an example with high confidence. We found that this trend obtained almost the same results on all validation examples, and we thus confirmed that the similarity is correlated to the incorrect rate (see Fig. 4 in the appendixes). Hence, we can assign appropriate weights to the margins computed with Eq.(4) by applying Eq.(12) to Eq.(10). For these reasons, we transform Eq.(10) to an appropriate representation with

$$s'(\boldsymbol{x}_i, y_i) = s(\boldsymbol{x}_i, y_i) \cdot c(\boldsymbol{x}_i, y_i). \tag{13}$$

### 3.4 INCORPORATION OF OUR APPROACH IN PREVIOUS IRAT METHODS

Above, we showed that the previous methods are insufficient, and we proposed *margin reweighting* to transform the margins computed by Eq.(4) to adequate representations with the top2 ratio. Here, we describe how to incorporate the proposed method into MAIL and WMMR.

First, we multiply the top2 ratio computed by Eq.(13) and the margins given by Eq.(4):

$$\tilde{m}(\boldsymbol{x}_i, y_i) = m(\boldsymbol{x}_i, y_i) \cdot s'(\boldsymbol{x}_i, y_i). \tag{14}$$

Then, we apply Eq.(14) to transform the margins to boost the importance of examples around the intersections of multi-class decision boundaries of multiple classes from weights functions of the previous methods:

$$\omega_{\text{WMMR}} = \exp(-\tilde{m}(\boldsymbol{x}_i, y_i)), \tag{15}$$
$$\omega_{\text{MAIL}} = \text{sigmoid}(-\gamma \cdot (\tilde{m}(\boldsymbol{x}_i, y_i) + \beta)). \tag{16}$$

Here, $\beta$ and $\gamma$ are hyperparameters, and in the experiment described below, we used the constants of the previous methods. Therefore, we only need to determine $\tau$ in the proposed method.

In addition, we omit the WMMR constant by replacing it with the proposed method, because it simply multiplies the margins computed by Eq.(4). We also warm up the model with $\omega = 1$ for

all examples during the initial training. In other words, the training within a set number of epochs is standard AT. This technique uses GAIRAT or MAIL, which aims to avoid inappropriate weight representations during the initial training. The training procedure incorporating the proposed method is detailed in Algorithm 1 in the appendixes.

## 4 EXPERIMENTS

This section demonstrates the performance obtained by incorporating the proposed method into WMMR and MAIL. In our experiment, we used the CIFAR-10/100 (Krizhevsky & Hinton, 2009) dataset to evaluate the standard accuracy for benign data and the adversarial robustness against several attacks. First, we describe the training settings and the details of how we evaluate the adversarial robustness in Section 4.1. Then, we report the quantitative evaluation results in Section 4.2 and discuss the hyperparameters and the necessity of the similarity computation in Section 4.3.

### 4.1 EXPERIMENTAL DETAILS

We used standard AT ("Standard") (Madry et al., 2018), WMMR, and MAIL as comparison methods. The evaluation results here consist of the average and standard deviation for five models trained with different random seeds. We also show the highest-performance model during training ("Best model") and at the end of training ("Last model"). The results for other methods and best models are shown in the appendixes to conserve space. For fair evaluation, the training settings with WRN34-10 on the CIFAR-10 dataset were integrated according to (Pang et al., 2021).

**Training Settings:**    We used ResNet-18 (He et al., 2016) and WideResNet34-10 (WRN34-10) (Zagoruyko & Komodakis, 2016) as base networks. We trained models during 120 epochs by using a batch size of 128 and SGD with a momentum of 0.9. The initial learning rates for each network were 0.01 (ResNet-18) and 0.1 (WRN34-10), and they were multiplied by 1/10 at $\{75, 90, 100\}$ epochs. The weight decays were set to $3.5 \times 10^{-3}$ (ResNet-18) and $0.5 \times 10^{-4}$ (WRN34-10). Finally, for all methods, we used PGD with a perturbation tolerance $\epsilon = 8/255$, a step size $\alpha = \epsilon/4$, and $N = 10$ steps for PGD.

**Hyperparameters:**    MAIL used $\beta = 0.5$ and $\gamma = 10$ for all models and datasets, while WMMR used $\alpha = 2$. These constants were the same both with and without the proposed method. The proposed method used $\tau = 2$ on both the CIFAR-10 and 100 datasets for WRN34-10, and $\tau = 2$ on CIFAR-10 and $\tau = 3$ on CIFAR-100 for ResNet-18. The amount of warmup was 75 epochs, after which we activated the weighted loss minimization.

**Adversarial Robustness:**    We evaluated not only the standard accuracy for benign examples ("Clean") but also the adversarial robustness against PGD-$K$ with $K = 100$ (PGD-100), PGD-100 with the loss function of CW, AutoPGD with the cross-entropy (APGD-CE) (Croce & Hein, 2020), and AutoAttack (AA) (Croce & Hein, 2020). As described in (Wang et al., 2021), we evaluated the performance for white-box attacks because black-box attacks are assumed to be easy to defend.

### 4.2 RESULTS

Tables 1 and 2 list the results on CIFAR-10 and on CIFAR-100, respectively.

**CIFAR-10:**  The results for ResNet-18, in the upper block of Table 1, showed improved adversarial robustness against PGD-100 and APGD-CE by incorporating the proposed method into MAIL (i.e., "Ours+MAIL"). Moreover, the "Clean" result showed somewhat improved performance. The results for incorporation into WMMR exhibited the same trend. Next, the results for WRN34-10, in the bottom block, showed better adversarial robustness against all attacks with the proposed method incorporated into WMMR (i.e., "Ours+WMMR"). In particular, the average robust accuracy against AA improved by 0.13, and the standard deviation for five models was significantly small. Meanwhile, "Ours+MAIL" showed comparable classification accuracy for benign examples, and it significantly improved the adversarial robustness against attacks in the PGD family, such as PGD-100 and APGD-CE.

Table 1: Performance of each method's "Last model" on CIFAR-10 (%). The upper block is for ResNet-18, and the bottom block is for WRN34-10. The best-performing method in each case in shown in bold, and ↑ indicates improved performance by incorporating the proposed method.

| | Clean | PGD-100 | CW-PGD | APGD-CE | AA |
|---|---|---|---|---|---|
| Standard | 85.69±0.15 | 48.86±0.22 | **49.26±0.20** | 48.34±0.23 | **46.38±0.26** |
| WMMR | **85.71±0.17** | 49.60±0.41 | 48.31±0.36 | 48.99±0.38 | 45.23±0.37 |
| MAIL | 82.53±0.30 | 56.66±0.17 | 47.71±0.26 | 55.75±0.13 | 45.22±0.31 |
| Ours + WMMR | 85.65±0.13 | 50.31±0.25↑ | 47.94±0.13 | 49.59±0.29↑ | 44.79±0.18 |
| Ours + MAIL | 82.66±0.10↑ | **57.96±0.13**↑ | 47.24±0.16 | **56.95±0.19**↑ | 44.78±0.29 |
| Standard | 87.89±0.17 | 48.11±0.50 | 49.33±0.45 | 47.66±0.47 | 46.82±0.38 |
| WMMR | **88.03±0.12** | 49.33±0.41 | 49.95±0.33 | 48.88±0.43 | 47.37±0.38 |
| MAIL | 86.40±0.13 | 59.30±0.21 | **51.82±0.22** | 58.57±0.26 | **49.29±0.14** |
| Ours+WMMR | 87.99±0.14 | 49.39±0.19↑ | 49.97±0.22↑ | 48.96±0.18↑ | 47.50±0.17↑ |
| Ours+MAIL | 86.48±0.24↑ | **60.56±0.42**↑ | 51.55±0.30 | **59.64±0.45**↑ | 48.92±0.30 |

Table 2: Performance of each model's "Last model" on CIFAR-100 (%). The upper block is for ResNet-18, and the bottom block is for WRN34-10. The best-performing method in each case is shown in bold, and ↑ indicates improved performance by incorporating the proposed method.

| | Clean | PGD-100 | CW-PGD | APGD-CE | AA |
|---|---|---|---|---|---|
| Standard | 60.10±0.22 | 28.65±0.21 | **27.89±0.17** | 28.12±0.18 | **25.01±0.13** |
| WMMR | 60.42±0.28 | 28.14±0.16 | 26.66±0.18 | 27.47±0.15 | 23.59±0.14 |
| MAIL | 56.77±0.19 | **31.23±0.08** | 25.68±0.26 | 30.45±0.09 | 22.99±0.09 |
| Ours + WMMR | **60.50±0.18**↑ | 28.24±0.12↑ | 26.96±0.20↑ | 27.58±0.14↑ | 23.86±0.15↑ |
| Ours + MAIL | 57.50±0.23↑ | **31.23±0.16** | 25.30±0.09 | **30.46±0.17**↑ | 22.61±0.10 |
| Standard | 62.63±0.33 | 24.82±0.12 | 25.80±0.17 | 24.53±0.13 | 23.70±0.13 |
| WMMR | 63.32±0.12 | 25.87±0.22 | 26.29±0.20 | 25.45±0.19 | 23.81±0.19 |
| MAIL | 62.56±0.21 | 34.29±0.08 | **29.31±0.19** | **33.58±0.09** | **26.50±0.16** |
| Ours+WMMR | **63.76±0.09**↑ | 25.45±0.24 | 26.51±0.15↑ | 25.14±0.21 | 23.94±0.18↑ |
| Ours+MAIL | 63.19±0.21↑ | **34.32±0.13**↑ | 28.94±0.18 | 33.47±0.18 | 26.18±0.14 |

**CIFAR-100:** Next, as seen the upper of Table 2 for ResNet-18, "Ours+MAIL" improved the accuracy for benign examples by 0.73, and the robustness against PGD-100 and APGD-CE was comparable. Meanwhile, "Ours+WMMR" achieved excellent performance not only in the "Clean" case but also for all attacks over 0.10. For WRN34-10, as seen in the bottom block of the table, "Ours+WMMR" improved the accuracy for benign examples by 0.44. In addition, it improved the adversarial robustness against both CW-PGD and AA, although the robustness against the PGD family slightly degenerated. In addition, "Ours+MAIL" improved the "Clean" case by 0.63, and the robustness against PGD-100 was comparable. Overall, these results showed that the proposed method was effective in improving the accuracy for benign examples, and that it obtained comparable or better adversarial robustness on the CIFAR-100 dataset.

To conserve space, the results for the "Best model" with each dataset and base network are given in Appendix E. In a nutshell, those results exhibited the same trend as the results given above for the "Last model".

### 4.3 ABLATION STUDY

Here, we consider the effect on performance when we eliminated the similarity of Eq.(11) or used different values of the hyperparameter $\tau$. We obtained the average and standard deviation for WRN34-10 with five models having different random seeds and the settings described in Section 4.1.

First, Table 3 lists the results of a performance comparison with $\tau = \{2, 3, 4\}$. The results on CIFAR-10, in the left block, showed that a large $\tau$ significantly increased the robustness against APGD-CE. Meanwhile, increased $\tau$ degraded the performance in the "Clean" and CW-PGD cases. In addition, a large $\tau$ increased the classical PGD robustness but degraded the AA robustness (see Appendix E).

Table 3: Ablation study for $\tau$. The left and right blocks are for CIFAR-10/100, respectively.

| | CIFAR-10 | | | CIFAR-100 | | |
|---|---|---|---|---|---|---|
| | Clean | CW-PGD | APGD-CE | Clean | CW-PGD | APGD-CE |
| $\tau = 2$ | **86.48±0.24** | **51.55±0.30** | 59.64±0.45 | 63.19±0.21 | **28.94±0.18** | **33.47±0.18** |
| $\tau = 3$ | 86.39±0.14 | 50.65±0.35 | 60.77±0.48 | 63.44±0.22 | 27.79±0.25 | 32.65±0.17 |
| $\tau = 4$ | 86.37±0.25 | 50.49±0.50 | **62.09±0.47** | **63.62±0.29** | 26.87±0.23 | 31.38±0.19 |

As seen in the right block of Table 3, the results on CIFAR-100 showed that a large $\tau$ increased the accuracy for benign examples but degraded the robustness against adversarial attacks. In particular, the robustness deteriorated by over 1.0 on CIFAR-10 and 2.0 on CIFAR-100. Thus, $\tau = 2$ was an appropriate choice for WRN34-10 on both datasets. The results for ResNet-18 are listed in Table 8 in Appendix E.

Next, Table 4 lists the results ("Clean", CW-PGD, and APGD-CE) when the similarity of Eq.(11) was eliminated from the proposed method. While the "Clean" and CW-PGD results deteriorated, the robustness against APGD-CE slightly improved. As seen in Table 7 in Appendix E, the robustness against PGD was comparable with the similarity eliminated, but the robustness against AA did not decrease. Although classical PGD is a strong attack, evaluation

Table 4: Ablation study for the "Last model" on CIFAR-10 for the similarity of Eq.(11). sim.: similarity.

| | w/ sim. | w/o sim. |
|---|---|---|
| Clean | **86.48±0.24** | 86.40±0.09 |
| CW-PGD | **51.55±0.30** | 51.07±0.23 |
| APGD-CE | 59.64±0.45 | **59.91±0.42** |

of the model performance for only PGD links overestimation of robustness. Therefore, for the proposed method, it is appropriate to use the similarity because of the comparable robustness against PGD despite improved the performance for CW-PGD or AA.

## 5 DISCUSSION AND LIMITATIONS

Surprisingly, standard AT achieved the best robustness against AA and CW-PGD in a small-capacity network such as ResNet-18. This result implies that MAIL is vulnerable to strong attacks when the network does not have enough capacity. Therefore, our approach may increase MAIL's innate vulnerability in exchange for dramatically improving its robustness against the PGD attack. Meanwhile, WMMR with the proposed method showed improved performance for almost all the attacks, including AA or CW-PGD, as compared with the original method, despite simply multiplying Eq.(13 by the margins. Instead of using a constant coefficient, our approach can apply different coefficients to the margins of each instance during training. These results show the necessity of transforming the margins to appropriate representations, because the margins transformed by our approach directly contribute to training without the constant coefficient.

Regarding the results on each dataset, the performance improvement on CIFAR-100 was limited in comparison to the improvement on CIFAR-10. Intuitively, our method should be an important role in far more classes with complex decision boundaries. Our method still just indirectly deals with the relations among multiple classes. Hence, inherent consideration of the multi-class margins may make networks robust on CIFAR-100 or larger-scale datasets, and we leave this as future work.

## 6 CONCLUSION

In this paper, we have revealed a weakness of previous IRAT methods, and we have proposed a novel metric to resolve this weakness. Unlike the previous methods, which define weights by considering only two probabilities, our approach successfully boosts the performance by incorporating the ratio of the top2 probability to the incorrect rate into the previous methods. However, the robustness improvement was limited on the CIFAR-100 dataset as compared with CIFAR-10. Image classification in the real world assumes more complex data and far more classes than in CIFAR-10/100. Hence, we will aim to improve the performance on large-scale datasets by carefully designing the weighting function.

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

## A    PROOF OF LEMMA 1

**Lemma 1.** *For the margin calculated by Eq.(4) and a training set $\mathcal{D} := \{(\boldsymbol{x}_i, y_i)\}_{i=1}^n$, there exist samples that satisfy*

$$m(\boldsymbol{x}_i, y_i) = m(\boldsymbol{x}_j, y_j). \tag{17}$$

*Proof:*

We divide the training dataset $\mathcal{D} = \{\boldsymbol{x}_i, y_i\}_{i=1}^n$ into the set of successfully classified samples, $\mathcal{S}^+ := \{(\boldsymbol{x}_i, y_i) \in \mathcal{D} \mid \arg\max p(\boldsymbol{x}_i) = y_i\}$, and the set of misclassified samples, $\mathcal{S}^- := \{(\boldsymbol{x}_i, y_i) \in \mathcal{D} \mid \arg\max p(\boldsymbol{x}_i) \neq y_i\}$. First, $\mathcal{S}^+$ and $\mathcal{S}^-$, Lemma 1 obviously holds when $p_{y_i}(\boldsymbol{x}_i) = p_{y_j}(\boldsymbol{x}_j)$. Next, we prove that Lemma 1 also holds for different probabilities, i.e., for $p_{y_i}(\boldsymbol{x}_i) \neq p_{y_j}(\boldsymbol{x}_j)$. Because all samples, including those in $\mathcal{S}^+$ and $\mathcal{S}^+$, absolutely satisfy $p_{y_i}(\boldsymbol{x}_i) > p_2(\boldsymbol{x_i})$ and $p_{y_i}(\boldsymbol{x}_i) < p_2(\boldsymbol{x_i})$, Lemma 1 must hold to satisfy the following equation:

$$|p_{y_i}(\boldsymbol{x}_i) - p_{y_j}(\boldsymbol{x}_j)| = |p_2(\boldsymbol{x}_i) - p_2(\boldsymbol{x}_j)|. \tag{18}$$

Thus, equal or approximately equal margins exist despite different true class and top2 probabilities. $\square$

## B    PROOF OF PROPOSITION 1

**Proposition 1.** *For input data $(\boldsymbol{x}_i, y_i)$ and $(\boldsymbol{x}_j, y_j)$, let $m(\boldsymbol{x}_i, y_i) = m(\boldsymbol{x}_j, y_j)$, $p_{y_i}(\boldsymbol{x}_i) > p_{y_j}(\boldsymbol{x}_j)$, and $p_{y_j}(\boldsymbol{x}_j) > \frac{1}{K}$. Then, the following inequality for the ratio of the top2 probability to the incorrect rate holds for each sample:*

$$o(\boldsymbol{x}_i, y_i) > o(\boldsymbol{x}_j, y_j), \;\; \text{where } o(\boldsymbol{x}, y) = \frac{p_2(\boldsymbol{x})}{\bar{p}_y(\boldsymbol{x})}. \tag{19}$$

*Proof:* Supposing that Lemma 1 holds under $p_{y_i}(\boldsymbol{x}_i) > p_{y_j}(\boldsymbol{x}_j)$ for each true class probability in the input data $(\boldsymbol{x}_i, y_i)$ and $(\boldsymbol{x}_j, y_j)$, the following inequality obviously holds for the top2 probabilities:

$$p_2(\boldsymbol{x}_i) > p_2(\boldsymbol{x}_j). \tag{20}$$

Hence, the incorrect rate for all data satisfies

$$\bar{p}_{y_i}(\boldsymbol{x}_i) < \bar{p}_{y_j}(\boldsymbol{x}_j). \tag{21}$$

Because we can guarantee that the top2 probability does not overshoot the incorrect rate in any case, the following inequality holds:

$$\frac{p_2(\boldsymbol{x}_i)}{\bar{p}_{y_i}(\boldsymbol{x}_i)} > \frac{p_2(\boldsymbol{x}_j)}{\bar{p}_{y_j}(\boldsymbol{x}_j)}, \tag{22}$$

Thus, we obtain $o(\boldsymbol{x}_i, y_i) > o(\boldsymbol{x}_j, y_j)$ by using $o(\boldsymbol{x}, y) = \frac{p_2(\boldsymbol{x})}{\bar{p}_y(\boldsymbol{x})}$. $\square$

## C    TRAINING PROCESS FOR PROPOSED ADVERSARIAL TRAINING

Algorithm 1 gives the training procedure for AT incorporating the proposed method. We activate the assigned weights for the loss after $\Omega$ epochs because the initial model is inappropriate for weighting. Hence, we use $\omega = 1$ (i.e., line 15) with less than $\Omega$ epochs. Moreover, the computed weights are multiplied by the coefficient $M$ instead of being directly applied to the loss. Despite the multiplication of $M$ by the weights in (Wang et al., 2021), Wang et al. (2021) do not discuss the necessity of this process. The experiments in this paper used $M = 3$.

---

**Algorithm 1** Adversarial training incorporating our approach

---

**Require:** Training dataset $\mathcal{D}$, batch size $n$, training epochs $T$, learning rate $\eta$, model parameter $\boldsymbol{\theta}$, amount of warm-up $\Omega$
**Require:** Function deriving adversarial perturbation $\mathcal{A}$
**Require:** Hyperparameters $\beta$, $\gamma$, $\tau$
1: **for** $t = 1, \ldots, T$ **do**
2:    **for** $\{\boldsymbol{x}_i, y_i | i = 1, \ldots, n\} \sim \mathcal{D}$ **do**
3:       $\hat{\boldsymbol{x}}_i \leftarrow \mathcal{A}(\boldsymbol{x}_i, y_i; \boldsymbol{\theta})$
4:       $s(\hat{\boldsymbol{x}}_i, y_i) = \tau - o(\hat{\boldsymbol{x}}_i, y_i)$
5:       $c(\hat{\boldsymbol{x}}_i, y_i) = \frac{<p(\hat{\boldsymbol{x}}_i), p'(\hat{\boldsymbol{x}}_i)>}{\|p(\hat{\boldsymbol{x}}_i)\|_2 \cdot \|p'(\hat{\boldsymbol{x}}_i)\|_2}$
6:       $s'(\boldsymbol{x}_i, y_i) = s(\boldsymbol{x}_i, y_i) \cdot c(\boldsymbol{x}_i, y_i)$
7:       $\tilde{m}(\hat{\boldsymbol{x}}_i, y_i) = m(\hat{\boldsymbol{x}}_i, y_i) \cdot s'(\hat{\boldsymbol{x}}_i, y_i)$
8:       **if** $T > \Omega$ **then**
9:         **if** MAIL **then**
10:           $\omega_i = \text{sigmoid}(\gamma \cdot (\tilde{m}(\hat{\boldsymbol{x}}_i, y_i) + \beta)) \times M$
11:         **else if** WMMR **then**
12:           $\omega_i = \exp(-\tilde{m}(\hat{\boldsymbol{x}}_i, y_i)) \times M$
13:         **end if**
14:       **else**
15:         $\omega_i = 1$
16:       **end if**
17:       model update:
18:       $\boldsymbol{\theta}_{t+1} \leftarrow \boldsymbol{\theta}_t - \eta \cdot \frac{1}{n} \sum_{i=1}^{n} \nabla_\theta \omega_i \cdot \ell(f_{\boldsymbol{\theta}_t}(\tilde{\boldsymbol{x}}_i), y_i)$
19:    **end for**
20: **end for**
21: **return** model parameter $\boldsymbol{\theta}$

---

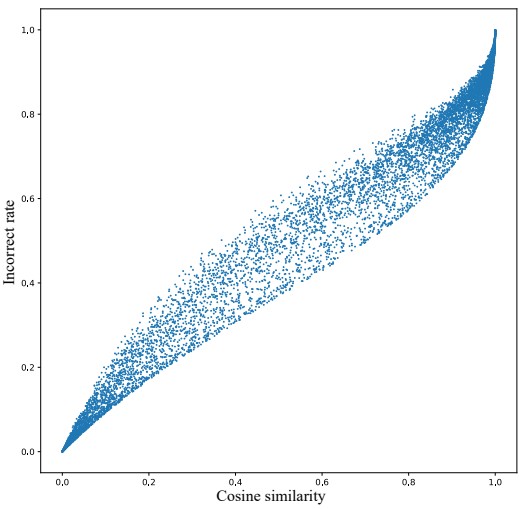

Figure 4: Similarity computed by Eq.(12) vs. the incorrect rate for WRN34-10 trained with standard AT on CIFAR-10. The two measures have an extremely large correlation, with a correlation coefficient of 0.98 computed by Eq.(23).

## D    CORRELATION BETWEEN INCORRECT RATE AND SIMILARITY

In our experiment, we computed the correlation between two measures for WRN34-10 trained by standard AT with the following equation:

$$r = \frac{\frac{1}{n} \sum_{i=1}^{n} (\bar{p}_{y_i}(\boldsymbol{x}_i) - \bar{p}')(c(\boldsymbol{x}_i, y_i) - c')}{\sqrt{\frac{1}{n} \sum_{i=1}^{n} (\bar{p}_{y_i}(\boldsymbol{x}_i) - \bar{p}')^2} \sqrt{\frac{1}{n} \sum_{i=1}^{n} (c(\boldsymbol{x}_i, y_i) - c')^2}}, \tag{23}$$

Table 5: Performance comparison results for the best models on CIFAR-10. The upper and lower blocks are for ResNet-18 and WRN34-10, respectively. The bold results indicate the best-performing method in each case, and the underlined results indicate successful robustness improvement by introducing the proposed method.

| | | Clean | FGSM | PGD-100 | CW-PGD | APGD-CE | AA |
|---|---|---|---|---|---|---|---|
| ResNet-18 | Standard | 83.84±0.171 | 57.71±0.17 | 50.98±0.21 | 50.50±0.31 | 50.55±0.22 | 47.86±0.13 |
| | GAIRAT | 83.63±0.06 | 61.13±0.35 | 56.62±0.46 | 37.72±0.66 | 53.37±0.47 | 34.76±0.66 |
| | EWAT | **83.97±0.52** | 57.41±0.51 | 50.56±0.31 | **50.51±0.50** | 50.14±0.34 | **48.03±0.48** |
| | WMMR | 83.74±0.23 | 58.05±0.16 | 52.33±0.21 | 49.12±0.12 | 51.70±0.25 | 46.57±0.12 |
| | MAIL | 82.37±0.23 | 60.41±0.08 | 56.99±0.20 | 47.90±0.19 | 56.07±0.23 | 45.44±0.18 |
| | Ours + WMMR | 83.87±0.27 | 58.37±0.17 | 52.91±0.16 | 48.52±0.24 | 52.17±0.16 | 46.04±0.24 |
| | Ours + MAIL | 82.66±0.09 | **61.74±0.25** | **58.36±0.30** | 47.43±0.23 | **57.24±0.30** | 44.82±0.36 |
| WRN34-10 | Standard | 86.67±0.21 | 61.33±0.15 | 54.17±0.23 | 54.05±0.15 | 53.79±0.22 | 51.73±0.32 |
| | GAIRAT | 85.74±0.90 | 63.50±0.38 | 60.45±0.18 | 45.85±2.48 | 57.29±0.85 | 43.12±2.42 |
| | EWAT | 85.93±0.46 | 60.98±0.62 | 54.17±0.24 | **54.27±0.19** | 53.79±0.25 | **51.91±0.16** |
| | WMMR | 86.57±0.18 | 61.30±0.23 | 55.16±0.37 | 52.76±0.39 | 54.64±0.35 | 50.36±0.33 |
| | MAIL | 86.28±0.12 | 63.60±0.02 | 59.77±0.21 | 51.91±0.29 | 58.93±0.26 | 49.47±0.33 |
| | Ours + WMMR | **86.91±0.31** | 61.36±0.32 | 55.00±0.25 | 53.27±0.08 | 54.49±0.30 | 50.73±0.12 |
| | Ours + MAIL | 86.45±0.19 | **64.66±0.14** | **61.05±0.25** | 51.87±0.17 | **60.08±0.21** | 49.26±0.22 |

Table 6: Performance comparison results for the best models on CIFAR-100. The upper and lower blocks are for ResNet-18 and WRN34-10, respectively. The bold results indicate the best-performing method in each case, and the underlined results indicate successful robustness improvement by introducing the proposed method.

| | | Clean | FGSM | PGD-100 | CW-PGD | APGD-CE | AA |
|---|---|---|---|---|---|---|---|
| ResNet-18 | Standard | 57.48±0.50 | 33.09±0.16 | 29.51±0.06 | **27.76±0.13** | 28.71±0.55 | **25.16±0.14** |
| | GAIRAT | 56.91±0.29 | 32.66±0.06 | 29.16±0.10 | 27.30±0.09 | 28.61±0.13 | 24.69±0.10 |
| | EWAT | 57.75±0.32 | 32.87±0.40 | 29.35±0.19 | 27.20±0.17 | 28.88±0.18 | 25.03±0.16 |
| | WMMR | 57.82±0.45 | 32.51±0.25 | 29.03±0.16 | 26.84±0.20 | 28.34±0.21 | 24.02±0.27 |
| | MAIL | 56.65±0.18 | **33.49±0.15** | **31.36±0.07** | 25.72±0.18 | **30.62±0.09** | 23.02±0.17 |
| | Ours + WMMR | **58.07±1.35** | 32.82±0.49 | 28.94±0.09 | 27.03±0.28 | 28.30±0.13 | 24.19±0.04 |
| | Ours + MAIL | 57.59±0.12 | 33.47±0.11 | 31.25±0.16 | 24.80±0.24 | 30.44±0.17 | 22.06±0.24 |
| WRN34-10 | Standard | 62.21±0.19 | 35.58±0.31 | 31.43±0.31 | **30.51±0.27** | 31.00±0.27 | **27.96±0.10** |
| | GAIRAT | 61.83±0.63 | 35.49±0.20 | 31.48±0.15 | 30.44±0.19 | 31.00±0.14 | 27.85±0.14 |
| | EWAT | 62.18±0.28 | 34.60±0.27 | 29.79±0.22 | 29.07±0.38 | 29.39±0.28 | 26.75±0.30 |
| | WMMR | 62.08±0.16 | 34.64±0.23 | 30.99±0.12 | 29.09±0.17 | 30.50±0.17 | 26.61±0.24 |
| | MAIL | 62.36±0.21 | **37.26±0.12** | **34.64±0.12** | 29.42±0.13 | **33.94±0.14** | 26.67±0.08 |
| | Ours + WMMR | 62.51±0.64 | 35.30±0.20 | 31.15±0.29 | 29.91±0.22 | 30.73±0.31 | 27.22±0.35 |
| | Ours + MAIL | **63.00±0.11** | 37.12±0.24 | 34.56±0.18 | 29.07±0.16 | 33.78±0.19 | 26.38±0.21 |

where $n$ is the number of samples, $\bar{p}' = \frac{1}{n}\sum_{i=1}^{n} \bar{p}_{y_i}(\boldsymbol{x}_i)$, and $c' = \frac{1}{n}\sum_{i=1}^{n} c(\boldsymbol{x}_i, y_i)$.

As shown in Fig. 4, there is a visual correlation between the two measures visually. Quantitatively, we observed a strong correlation with $r = 0.98$. Therefore, we can mitigate inappropriate behavior of the top2 ratio for correct samples with high confidence by relying on the similarity.

# E  ADDITIONAL RESULTS

This section describes experimental results that are omitted from the main paper to conserve space. The experimental settings were the same as those described in Section 4.1. Additionally, we added GAIRAT and EWAT as comparison methods and FGSM as an attack.

Tables 5 and 6 list the adversarial robustness and classification accuracy for the best model results of each method on CIFAR-10/100, respectively. The best model in the experiment was the checkpoint that achieved the best robustness against PGD-20 with the validation data. In terms of adversarial robustness, we improved the performance for the standard accuracy and certain attacks by 0.32-0.58 (for ResNet-18) and 0.06-0.51 (for WRN34-10) with "Ours+WMMR". "Ours+MAIL" improved the performance for the standard accuracy and certain attacks by 1.17-1.37 (for ResNet-18) and 1.06-

Table 7: Ablation study for the similarity of Eq.(11) on CIFAR-10. The upper and lower blocks are for the last and best models, respectively. sim.: similarity.

|  |  | Clean | FGSM | PGD-100 | CW-PGD | APGD-CE | AA |
|---|---|---|---|---|---|---|---|
| Last | w/ sim. | **86.48±0.24** | 64.52±0.20 | 60.56±0.42 | **51.55±0.30** | 59.64±0.45 | **48.92±0.30** |
|  | w/o sim. | 86.40±0.09 | 64.52±0.28 | **60.88±0.36** | 51.07±0.23 | **59.91±0.42** | 48.73±0.23 |
| Best | w/ sim. | **86.45±0.19** | 64.66±0.14 | 61.05±0.25 | **51.87±0.17** | 60.08±0.21 | **49.26±0.22** |
|  | w/o sim. | 86.27±0.17 | **64.88±0.22** | **61.35±0.17** | 51.45±0.21 | **60.39±0.17** | 48.77±0.26 |

Table 8: Ablation study for the hyper-parameter $\tau$ with ResNet-18, for the last model. The upper and lower blocks are for CIFAR-10 and CIFAR-100, respectively.

|  |  | Clean | FGSM | PGD-100 | CW-PGD | APGD-CE | AA |
|---|---|---|---|---|---|---|---|
| CIFAR-10 | $\tau = 2$ | 82.66±0.10 | 61.46±0.20 | 57.96±0.13 | **47.24±0.16** | 56.95±0.19 | **44.78±0.29** |
|  | $\tau = 3$ | **82.78±0.16** | 61.52±0.47 | 57.81±0.49 | 47.07±0.18 | 56.71±0.48 | 44.54±0.30 |
|  | $\tau = 4$ | 82.35±0.32 | **63.66±0.19** | **60.64±0.21** | 45.54±0.39 | **59.30±0.20** | 42.64±0.39 |
| CIFAR-100 | $\tau = 2$ | 56.80±0.26 | 33.43±0.14 | 31.14±0.18 | **25.81±0.13** | 30.36±0.14 | **23.09±0.16** |
|  | $\tau = 3$ | 57.50±0.23 | **33.47±0.11** | **31.23±0.16** | 25.30±0.09 | **30.46±0.17** | 22.61±0.10 |
|  | $\tau = 4$ | **57.78±0.16** | 33.38±0.08 | 30.93±0.05 | 24.68±0.11 | 30.15±0.06 | 21.83±0.05 |

1.28 (for WRN34-10). Surprisingly, while the WRN34-10 results for GAIRAT were comparable or better than those for MAIL, "Ours+MAIL" outperformed GAIRAT.

Focusing on the results for CIFAR-100 in Table 6, we see that "Ours+MAIL" had comparable or degraded performance relative to MAIL against certain attacks. Meanwhile, "Ours+MAIL" improved the "Clean" case by 0.94 (ResNet-18) and 0.64 (WRN34-10). "Ours+WMMR" improved the performance against many attacks and outperformed the "Clean" accuracy. The WRN34-10 results showed improved robustness against all attacks; in particular, "Ours+WMMR" improved the robustness by 0.82 at best against CW-PGD and AA.

Table 7 lists the performance results for the best and last models with and without the similarity on CIFAR-10. As described in Section 4.3, use of the similarity not only improved the robustness against CW-PGD and AA but also improved the "Clean" accuracy. The same results were obtained for the best model.

Table 8 lists the results for ResNet-18 trained with different $\tau$ values, i.e., $\tau = \{2, 3, 4\}$. With increasing $\tau$, while the robustness against CW-PGD and AA significantly deteriorated, the robustness against PGD and FGSM was dramatically improved on CIFAR-10. On CIFAR-100, a large $\tau$ improved the "Clean" accuracy, while the PGD and FGSM results were comparable. Meanwhile, the robustness against CW-PGD and AA deteriorated on both datasets. From these results, we determined the best values as $\tau = 2$ for CIFAR-10 and $\tau = 3$ for CIFAR-100.

