# OpenReview forum: "Revisiting Instance-Reweighted Adversarial Training"
_ICLR.cc/2023/Conference — Submitted to ICLR 2023_

### Official Review · Reviewer_QLpw · 2022-10-22

**Confidence:** 4
**Correctness:** 3
**Technical Novelty And Significance:** 3
**Empirical Novelty And Significance:** 2
**Recommendation:** 3

**Clarity, Quality, Novelty And Reproducibility:**

Clarity: Overall, the presentation is clear.

Quality: The main concern of this work is the result. The gain seems to be dominated by the variance and the proposed method has very minor contribution to the gain.

Originality: The paper is novel from my best understanding.

Reproducibility: The author does not provide the code in the submission. There is no clear algorithm section in the paper where the reader can refer, so it might not be simple to fully reproduce the work.


**Strength And Weaknesses:**

Strength
(1).	Figure 1 is a good illustration of the overall idea of the paper.
(2).	The motivation of this work is interesting.
(3).	The variance of the results is reported in the table.

Weaknesses
(1).	The gain of the proposed method is very minor, as shown in Table 1 and Table 2. Take the bottom block of Table 1 for example. When comparing WMMR with Ours+WMMR, we can see that the proposed method is almost the same as the prior work (some of them is even worse). This conclusion also holds both Cifar10 and Cifar100 dataset (Table 1 and Table 2). This conclude that the proposed method is not working. The author is suggested to provide more convincing results to support the effectiveness of the proposed method.
(2).	Table 3 shows completely different trend when \tau increases from 2 to 4. For APGD-CE, the accuracy of Cifar10 improves when \tau increases, while the accuracy of Cifar100 degrades when \tau increases. This shows that the conclusion is inconsistent between 2 datasets. The author does not provide explanation to this circumstance.
(3).	Table 4 shows that there is no much difference with or without the similarity of Eq.(11) and it seems that the difference is coming from the variance


**Summary Of The Paper:**

This paper aims to address the problem of robust overfitting in the adversarial training, where the robust model overfits to the adversarial examples. Prior works address this by assigning different weight to different instances according to their importance. However, this has not been well extended to multi-class classification scenario, where the instance could be closed to the intersection of multiple class boundaries. In this work, larger weight is assigned to these examples and better suit for multi-class classification. However, the experiment does not demonstrate the effectiveness of the proposed method.

**Summary Of The Review:**

The main concern of this work is the performance. There is little evidence that the proposed method is improving over the prior works.

---

### Official Review · Reviewer_UgLM · 2022-10-22

**Confidence:** 5
**Correctness:** 3
**Technical Novelty And Significance:** 2
**Empirical Novelty And Significance:** 1
**Recommendation:** 3

**Clarity, Quality, Novelty And Reproducibility:**

The clarity, quality and reproducibility is OK. Although the authors did not strictly follow the experimental settings of the baselines, the reproduction is very simple. The proposed method is mainly based on some improvements of the existing technology and the novelty is limited.

**Strength And Weaknesses:**

+The method is well organized and extensive evidence is provided to support the proposed method.

+The experiments are sufficient, including multiple benchmark datasets and different network structures.

-The motivation is somewhat unclear. What is the main difference between the proposed margin reweighting method and the baseline? Why do instances closer to the intersection of the decision boundaries require larger weight values? How does the instance close to the intersection of the decision boundaries with a large weight value improve robustness?

-The experimental results are pessimistic. AA is currently the most reliable robustness evaluation method. The proposed method has no improvement in AA, but it has declined compared to baselines. Therefore, I am concerned that the method proposed in this paper has no substantial robustness improvement.

-There are some vague statements, such as:
“Although GAIRAT achieves better robustness than standard AT, it is vulnerable against attacks other than PGD because it defines the importance in terms of the LPS.”
First, AA is currently the most reliable robustness evaluation method, and the robustness of GAIRAT is much lower than that of standard AT. Second, CE loss is invariant to shifts of the logits but not to rescaling [1,2]. Is the robustness of GAIRAT significantly reduced due to the amplitude of its scaling? I am not quite sure what “defines the importance in terms of the LPS” means.

[1] Hitaj, Dorjan, et al. "Evaluating the robustness of geometry-aware instance-reweighted adversarial training." arXiv preprint arXiv:2103.01914 (2021).

[2] Croce, Francesco, and Matthias Hein. "Reliable evaluation of adversarial robustness with an ensemble of diverse parameter-free attacks." International conference on machine learning. PMLR, 2020.


**Summary Of The Paper:**

This paper finds that previous methods, which computed margins using the true class and the most confusing class, are insufficient for handling the instance reweighting in the multi-class cases. The authors argue that instances close to the intersection of the decision boundaries should be more important, so they should have a larger margin weight. Therefore, the authors propose an improved margin reweighting method and verify the effectiveness on multiple datasets and network structures.

**Summary Of The Review:**

My main concern is that the proposed method cannot substantially improve the adversarial robustness of the network.

---

### Official Review · Reviewer_mrgM · 2022-10-24

**Confidence:** 5
**Correctness:** 3
**Technical Novelty And Significance:** 2
**Empirical Novelty And Significance:** Not applicable
**Recommendation:** 3

**Clarity, Quality, Novelty And Reproducibility:**

**Clarity:** The paper is clear and easy to understand. However, the paper explains their approach with a redundant equation for their simple approaches which weighs more on the incorrect rate based on the margin equation that is suggested in the previous works.

**Quality:** The presentation of the paper seems fine. However, since the proposed approach could be adjusted in WMMR and MAIL, I hope the authors rearrange rows of table 1 and table 2 as follows which could show the performance gain more effectively. (Standard / WMMR / WMMR + Ours / MAIL / MAIL + Ours)

**Novelty:** The idea and approach seem has a lack novelty. Since the previous works already suggest the main idea of instance-reweighting in adversarial training based on the margin, additional weights (incorrectness) on the previous equation seem highly related to me.

**Reproducibility:** The paper has well reproducibility which elaborates well on the details.

**Strength And Weaknesses:**

**Strength**
- This paper tackles the limitation of previously suggested margin-based adversarial training which did not consider the multi-class margins.
- The paper is easy to understand, clear, and well-written.

**Weakness**
- It is quite difficult to understand what is the motivation for using cosine similarity weights on top of equation 10. Especially, it is unclear why the suggested weight equation could be a better standard for instance reweighted adversarial training.
- Based on the performance, I am not sure it is adequate to claim that the suggested weight equation helps to improve the robustness against AutoAttack. In ResNet18 and CIFAR10/CIFAR100, reweighting does not show any effectiveness. In larger architecture, the proposed method could not improve the previous margin based reweighting both in CIFAR10 and CIFAR100.
- If the margin equation is the limitation to representing the multi-class margins, it seems better to use entropy as a standard.

**Minor**
- There is no explanation for M in equation 12.

**Summary Of The Paper:**

This paper proposes instance-reweighted adversarial training (IRAT) which weighs the instances based on the suggested standard in the adversarial loss term. This paper suggests the margin-based standard which weighs more when the instance is near the decision boundary. While the previous margin-based instance-reweighted adversarial training approaches are already proposed, previous works have limitations to represent multi-class margins with suggested margin formulation. Therefore, the current paper proposes a modified margin equation that considers the multi-class.

**Summary Of The Review:**

This paper suggests the modified margin-based reweighting scheme for adversarial training which shows a lack of novelty.
Further, in the results, it is difficult to find the effectiveness of the proposed methods.
Overall, I recommend rejecting this paper.

---

### Official Review · Reviewer_tft5 · 2022-10-26

**Confidence:** 4
**Clarity, Quality, Novelty And Reproducibility:** The paper is clearly written by addre…
**Correctness:** 3
**Technical Novelty And Significance:** 3
**Empirical Novelty And Significance:** Not applicable
**Recommendation:** 5

**Strength And Weaknesses:**

Pros:
1.	The idea is novel and it solves a challenge of traditional instance-weighted adversarial training for multi-class classification problems.
2.	The presentation of the paper is clear and the method is easy to follow.

Cons:
The problem with this paper is that experiments are not so convincing.
1. I think results on CIFAR-10 and CIFAR-100 datasets show that there is only marginal improvement in some cases. This might show that the challenge is not so significant for multi-class classification problems.
2. Only about half of the methods improve performance when combined with the proposed method. For the strongest AA attack, the proposed method seems only achieve minor improvement. And it seems to be difficult to explain why others do not work.



**Summary Of The Paper:**

This paper proposed a novel method for adversarial training called marginal reweighting. Previous instance-weighted adversarial training method does not take other classes into account for multiple classes classification. The key challenge is to quantify the importance of margins. This paper uses cosine similarity to represent the importance of the incorrect rate by leveraging the predicted probability distributions. And it takes all other classes into account and it could be incorporated into IRAT methods. It conducts experiments on CIFAR datasets and ablation studies to show the effectiveness of the proposed method.

**Summary Of The Review:**

I would like to see more convincing results on larger datasets or more valuable experimental settings like problems with more classes. We would like to increase the scores if more convincing experiments are provided.

---

### Decision · Program_Chairs · 2023-01-20

**Decision:**

Reject

**Justification For Why Not Higher Score:**

This paper targets a realistic problem and designs a reweighting algorithm to handle it. However, as mentioned, the three-fold weaknesses make it below acceptance. Reviewers raise several concerns about this paper. The authors do not provide responses to the concerns in the rebuttal period.

**Justification For Why Not Lower Score:**

N/A

**Metareview: Summary, Strengths And Weaknesses:**

**Summary**

This paper focuses on robust overfitting in adversarial training, where deep models overfit to those adversarial examples. A margin reweighting method is proposed, which assigns larger weights to examples close to the intersection of decision boundaries. Experiments on multiple benchmarks confirm the effectiveness of the proposed method.


**Strengths**

- The research problem is interesting and practical.
- Both the writing and organization of this paper are overall great.

**Weaknesses**
- The design of the method needs to be further clarified, e.g., cosine similarity weights.
- The novelty and contributions of this paper are unclear. The targeted problem has been explored by previous works. Additionally, the idea of reweighting was studied. This paper did not provide a comprehensive discussion of relevant literature.
- Experimental results are not convincing. In lots of cases, the performance improvement by the proposed method is marginal.